# A New Principle toward Robust Matching in Human-like Stereovision

**DOI:** 10.3390/biomimetics8030285

**Published:** 2023-07-02

**Authors:** Ming Xie, Tingfeng Lai, Yuhui Fang

**Affiliations:** School of Mechanical and Aerospace Engineering, Nanyang Technological University, Singapore 639798, Singapore; lait0012@e.ntu.edu.sg (T.L.); fang0119@e.ntu.edu.sg (Y.F.)

**Keywords:** visual signals, stereovision, image sampling, feature extraction, incremental learning, match-maker, cognition, recognition, possibility function

## Abstract

Visual signals are the upmost important source for robots, vehicles or machines to achieve human-like intelligence. Human beings heavily depend on binocular vision to understand the dynamically changing world. Similarly, intelligent robots or machines must also have the innate capabilities of perceiving knowledge from visual signals. Until today, one of the biggest challenges faced by intelligent robots or machines is the matching in stereovision. In this paper, we present the details of a new principle toward achieving a robust matching solution which leverages on the use and integration of top-down image sampling strategy, hybrid feature extraction, and Restricted Coulomb Energy (RCE) neural network for incremental learning (i.e., cognition) as well as robust match-maker (i.e., recognition). A preliminary version of the proposed solution has been implemented and tested with data from Maritime RobotX Challenge. The contribution of this paper is to attract more research interest and effort toward this new direction which may eventually lead to the development of robust solutions expected by future stereovision systems in intelligent robots, vehicles, and machines.

## 1. Introduction

We are living inside an ocean of signals. Among all the signals, visual signals should be the ones with the upmost importance. This is because without the visual signals, human beings will not be able to undertake many activities in the physical world. Similarly, visual signals are extremely important to today’s autonomous robots, vehicles and machines [1]. Hence, research works on enabling robots, vehicles and machines to gain human-like intelligence from the use of visual signals should never be undermined or shadowed by the development of alternative sensors such as Radar [2] and LiDAR [3]. 

In this paper, we present a new principle which addresses the most difficult problem in stereovision, which is to achieve stereo matching as robust as possible [4]. The motivation behind our research works comes from projects dedicated to the development of intelligent humanoid robots [5] as well as autonomous surface vehicles [6], as shown in Figure 1.

For both platforms in Figure 1, their intelligence and autonomy greatly depend on the outer loop which consists of perception, planning and control. Among all possible modalities of doing perception, visual perception is a very important one. Especially, the goal toward achieving human-like visual perception must start with the use of binocular vision or stereovision [7]. Hence, research on human-like stereovision should be a non-negligible topic in both artificial intelligence and robotics. Therefore, the purpose of this paper is to present a new principle which advances the current state of the art in developing human-like stereovision for autonomous robots, vehicles, and machines [8].

The remaining part of the paper is organized as follows: Section 2 outlines the biggest challenge faced by stereovision. Section 3 briefly discusses similar works dedicated to stereo matching. Section 4 describes the proposed new principle which addresses stereo matching problem. Section 5, Section 6, Section 7, Section 8, Section 9 and Section 10 present the details of the key steps inside the proposed principle. Section 11 shows some preliminary results. The conclusions are given in Section 12.

## 2. Problem Statement

Stereovision enables human beings to classify entities, to identify entities, and to localize entities. Clearly, stereovision is a very powerful system or module which provides answers to the following questions:What are the classes of perceived entities?What are the identities of perceived entities?Where are the locations of perceived entities inside related images?Where are the locations of perceived entities inside scenes?

A complete solution, which could fully answer the above questions, actually depends on the availability of the working principles underlying image sampling (NOTE: this is a largely overlooked sub-topic), entity cognition (NOTE: vaguely named as deep learning in literature [9]), entity recognition (NOTE: vaguely named as object detection in literature [10]), and entity matching [11] as shown in Figure 2, etc. 

As illustrated in Figure 2, when an entity is placed at location Q in a scene, its stereo images will appear at location a in left image and location b in right image, respectively. Then, the stereo matching problem could be stated as follows:

Given the location of an entity seen by the left camera, how to determine the location of the same entity seen by the right camera? [12].

The above problem implies the following two challenges:How to determine the presence of an entity in the left camera’s image plane?How to find the match in the right camera’s image plane if an entity has been detected in the left camera’s image plane?

It is important to further pay attention to the root causes behind these two challenges. The major root causes include [13]:Variations of entities in sizeVariations of entities in orientationVariations of entities’ images due to lighting conditionsVariations of entities’ images due to occlusions as shown in Figure 3aVariations of entities’ images due to image sampling process as shown in Figure 3b

In practice, we could cope with the issues raised by the variations of images in sizes and orientations if we could afford to have enough computational powers allocated to process images at multiple scales and multiple rotations. Also, we could cope with the variations of lighting conditions if we could make use of cameras with built-in functions of automatic illumination compensation and/or automatic contrast equalization. Hence, our remaining effort should focus on dealing with the issue raised by the occurrence of partial views faced by stereovision.

## 3. Similar Works on Stereo Matching

Stereo matching is an old problem in computer vision. In literature, there is a tremendous amount of works dedicated to solving the problem faced by stereo matching. For example, there are:Methods which make the attempt of matching points within a pair of stereo images [14].Methods which make the attempt of matching edges or contours within a pair of stereo images [15].Methods which make the attempt of matching line segments within a pair of stereo images [16].Methods which make the attempt of matching curves within a pair of stereo images [17].Methods which make the attempt of matching regions within a pair of stereo images [18].Methods which make the attempts of matching objects within a pair of stereo images [19].

The proposed principle in this paper falls into the category of making the attempt of matching entities within a pair of stereo images. Here, an entity may broadly refer to an object, a person, an animal, a building, or a machine, etc. In literature, the existing solutions in this category focus on the use of deep convolution to do feature extraction which is then followed by the use of artificial neural network to do tuning and prediction. Such methods simply depend on the process of bottom-up optimization (e.g., back propagation algorithm) and the use of features extracted in time-domain. 

In contrast, our proposed principle advocates the use of top-down design process in which we promote the use of hybrid features (i.e., features from both time-domain and frequency domain) as well as the use of the improved version of RCE neural network [20]. RCE neural network [21,22,23], which was discovered in 1970s by a research team led by a laureate [23] of 1972’s Nobel prize in Physics, is fundamentally different from artificial neural network. So far, to the best of our knowledge, there is no other better way of designing human-like cognition and recognition than the use of RCE neural network or its improved versions [20,21,22,23].

It is worth acknowledging that despite the huge amount of research works dedicated to stereovision, the achieved results are far behind the performance of human beings’ stereovision. Obviously, we should not stop the continuous investigation which aims at looking for better principles of, or solutions to, human-like stereovision.

## 4. The Outline of Proposed Principle

Human vision is attention-driven in a top-down manner. The attention could be triggered by the occurrences of reference entities such as appearances of persons, appearances of animals, appearances of objects, appearances of machines, appearances of geometries (e.g., lines, curves, surfaces, volumes, etc.), appearances of photometry (e.g., chrominance and luminance, etc.), appearances of textures, etc. Such reference entities could be learnt by a cognition process incrementally in real-time. However, the occurrences of familiar reference entities should be the responses of an internal recognition process.

Inspired by the innate processes of human vision, we propose a new principle which imitates the attention-driven behavior of human vision. The main idea of the proposed new principle is outlined in Figure 4.

Without loss of generality, we assume that the attention is to be recognized from the video streams of the left camera. The key steps involved in the proposed new principle include:Image acquisition by both cameras.Image sampling on video stream from left camera.Hybrid feature extraction for each image sample.Cognition of image samples if they correspond to the training data of reference entities inside training images.Recognition of image samples if they correspond to the possible occurrences of reference entities inside real-time images.Forward/Inverse processes of template matching, which work together so as to find the occurrence of matched candidate in the right image, if a recognized entity is present in the left image.

In the subsequent sections, we will describe the details of key steps 2 to 6.

## 5. Top-Down Strategy of Doing Image Sampling

An image may contain many entities of interest. One of the biggest challenges faced by image understanding or image segmentation/grouping is to divide an image into a matrix of image samples, each of which just contains the occurrence or appearance of a single entity. In theory and in practice, there is no solution which could generally guarantee such results expected by the subsequent visual processes in stereovision.

In addition, the problem of finding better ways to do image sampling did not receive enough attention in the research community. One major reason is because many people believe that it is good enough to use a sub-window to scan an input image so as to obtain all the possible image samples. However, this way of doing image sampling has serious drawbacks such as:It is difficult to determine, or to justify, the size of sub-window which is used to scan an input image. If the size of sub-window is allowed to be dynamically changed, then the next question is how to do such dynamic adjustment of sizes.The number of obtained image samples is independent of the content inside an input image. For example, an input image may contain a single entity. In this case, the scanning method will still produce many image samples which will be the input to subsequent visual processes of classification, identification, and grouping, etc. Obviously, irrelevant image samples may potentially cause troubles to these visual processes of recognition.

In this paper, we advocate a top-down strategy which iteratively divides an input image into a list of sets which contain linearly growing numbers of image samples of different sizes. If we denote $S_{k}$ a set which contains k image samples, one way to obtain $S_{k}$ is to uniformly divide an input image into a matrix of $d_{v}\times d_{h}$ samples, in which $d_{v}\times d_{h}=k$. For example, if we iteratively divide an input image into:
$S_{k}$ with one sample, then k=1 and $d_{v}\times d_{h}\in [1\times 1]$.$S_{k}$ with two samples, then k=2 and $d_{v}\times d_{h}\in [1\times 2,2\times 1]$.$S_{k}$ with three samples, then k=3 and $d_{v}\times d_{h}\in [1\times 3,3\times 1]$.$S_{k}$ with four samples, then k=4 and $d_{v}\times d_{h}\in [1\times 4,4\times 1,2\times 2]$.and so on.


This top-down strategy of doing image sampling is suitable for both parallel implementation and sequential implementation. 

Before sending the image samples to the next visual process of extracting features, it is necessary to normalize the size of image samples so as to make them to be comparable in size. In practice, it is trivial to scale up or down the size of an image sample to any chosen standard value. By now, we could represent image sample set $S_{k}$ as follows:(1)$$S_{k}=\left\{I_{j,r}\left(u,v\right),I_{j,g}\left(u,v\right),I_{j,b}\left(u,v\right),u\in \left[0,U-1\right],v\in \left[0,V-1\right],j\in \left[1,k\right]\right\}$$
where (r,g,b) are the three primary color components at index coordinates (u,v) inside set $S_{k}$’s jth image sample $I_{j}(u,v)$ which has the size of V×U. Hence, by default, each image acquisition module in stereovision outputs color images, each of which is represented by a set of three matrices such as $I_{j,r}\left(u,v\right),I_{j,g}\left(u,v\right),I_{j,b}\left(u,v\right)$ in Equation (1). 

## 6. Feature Extraction from Sample Image in Time-Domain

Mathematically speaking, the periodicity in space is equivalent to the periodicity in time. Hence, without loss of generality, we consider the spatial axes of an image or image sample as time axes. In this way, we could focus our discussions on how to extract features in time domain as well as in frequency domain.

Feature extraction in time domain has been extensively investigated by the research community of image processing and computer vision. In general, the basic operations include the computations of nth order derivatives where n could be equal to 0, 1, 2, 3, and any other larger value of integer. Here, the zero-order derivatives could refer to the results obtained by the operation of image smoothing for noise reduction (e.g., to use Gaussian filters). 

In literature, there are also many advanced studies which explore the use of Laplacian filters, Gabor filters, Wavelet filters, Moravec corner filter, Harris-Stephens corner filter, and Shi-Tomasi corner filter, etc. Hence, feature extraction in time domain is a very rich topic.

From the results shown in Figure 5, it is clear to us that higher order derivatives do not significantly provide extra information. The data of zero-order derivatives and first-order derivatives should be good enough for us to extract meaningful features in time domain. 

In practice, the zero-order derivatives could be obtained by convoluting set $S_{k}$’s jth image sample $I_{j}(u,v)$ with a discrete Gaussian filter such as:(2)$$G\left(u,v\right)=\frac{1}{16}\left[\begin{array}{ccc} 1 & 2 & 1\\ 2 & 4 & 2\\ 1 & 2 & 1 \end{array}\right],v\in \left[0,2\right],u\in [0,2]$$

If we represent the results (i.e., a matrix of zero-order derivatives of all the color components) of zero-order derivatives as follows:(3)$$S_{k,0}=\left\{I_{j,r_{0}}\left(u,v\right),I_{j,g_{0}}\left(u,v\right),I_{j,b_{0}}\left(u,v\right),u\in \left[0,U-1\right],v\in \left[0,V-1\right],j\in \left[1,k\right]\right\}$$

Then, the first-order derivatives could be obtained by convoluting each matrix in $S_{k,0}$ with the following two Sobel filters: (4)$$C_{h}\left(u,v\right)=\left[\begin{array}{ccc} +1 & +2 & +1\\ 0 & 0 & 0\\ -1 & -2 & -1 \end{array}\right],v\in \left[0,2\right],u\in [0,2]$$
and
(5)$$C_{v}\left(u,v\right)=\left[\begin{array}{ccc} +1 & 0 & -1\\ +2 & 0 & -2\\ +1 & 0 & -1 \end{array}\right],v\in \left[0,2\right],u\in [0,2]$$

Clearly, the convolution with filter in Equation (4) will result in the horizontal components of the first-order derivatives while the convolution with filter in Equation (5) will result in the vertical components of the first-order derivatives. The $L^{2}$ norms computed from these two components will yield the results of the first-order derivatives of image samples $S_{k,0}$, which could be represented by:(6)$$S_{k,1}=\left\{I_{j,r_{1}}\left(u,v\right),I_{j,g_{1}}\left(u,v\right),I_{j,b_{1}}\left(u,v\right),u\in \left[0,U-1\right],v\in \left[0,V-1\right],j\in \left[1,k\right]\right\}$$

Therefore, for image sample *j* in set $S_{k}$, it actually has six image matrices which are: $\left[I_{j,r_{0}}\left(u,v\right),I_{j,g_{0}}\left(u,v\right),I_{j,b_{0}}\left(u,v\right)\right]$ in Equation (3) and $\left[I_{j,r_{1}}\left(u,v\right),I_{j,g_{1}}\left(u,v\right),I_{j,b_{1}}\left(u,v\right)\right]$ in Equation (6). Then, the next question is how to determine a feature vector $F_{j}$ which meaningfully represents image sample *j* in set $S_{k}$.

A simple answer to the above question could be to convert the six image matrices of a sample into their vector representations (i.e., a 2D matrix is re-arranged as a 1D vector). Then, by putting these six image vectors together, we will obtain feature vector $F_{j}$. The advantage of this method is its simplicity. However, the noticeable drawback is the large dimension of feature vector $F_{j}$. Then, we may want to know whether there is a better way of determining feature vector $F_{j}$ from image matrices, or not. 

So far, there is no theoretical answer to this question. Maybe, a practical way is to design workable solutions which could be suitable for applications in hands. In this way, a library of workable solutions may empower autonomous robots, vehicles, or machines to adapt their behaviors to real-time situations or applications. Clearly, this topic still offers opportunities for further or continuous research works.

Here, we propose a simple and practical way of determining feature vector $F_{j}$ from image matrices. The idea is to compute statistics from a set of image matrices. Interestingly, the two obvious types of statistics are the mean values and standard deviations.

For example, if $\left\{I\left(u,v\right),u\in \left[0,U-1\right],v\in \left[0,V-1\right]\right\}$ is an image matrix of single values such as red components, green components, blue components, or their individual first-order derivatives, each value in $\left\{I\left(u,v\right),u\in \left[0,U-1\right],v\in \left[0,V-1\right]\right\}$ could be considered as a kind of measurement of approximate electromagnetic energy. Therefore, we could compute the following four meaningful statistics from $\left\{I\left(u,v\right),u\in \left[0,U-1\right],v\in \left[0,V-1\right]\right\}$, which are:
The mean value $I_{a}$ of approximate electromagnetic energy: (7)$$I_{a}=\frac{1}{UV}\sum_{v=0}^{V-1}\sum_{u=0}^{U-1}I(u,v)$$The square-root of the variance $\sigma_{I}$ of approximate electromagnetic energy:(8)$$\sigma_{I}=\sqrt{\frac{\sum_{v=0}^{V-1}\sum_{u=0}^{U-1}{\left(I\left(u,v\right)-I_{a}\right)}^{2}}{UV}}$$The horizontal distribution $\sigma_{u}$ of approximate electromagnetic energy:(9)$$\sigma_{u}=\sqrt{\frac{\sum_{v=0}^{V-1}\sum_{u=0}^{N-1}I(u,v)\times {\left(u-u_{c}\right)}^{2}}{\sum_{v=0}^{V-1}\sum_{u=0}^{U-1}I(u,v)}}$$
with:(10)$$u_{c}=\frac{\sum_{v=0}^{V-1}\sum_{u=0}^{N-1}\left\{I(u,v)\times u\right\}}{\sum_{v=0}^{V-1}\sum_{u=0}^{U-1}I(u,v)}$$
and
(11)$$v_{c}=\frac{\sum_{v=0}^{V-1}\sum_{u=0}^{N-1}\left\{I(u,v)\times v\right\}}{\sum_{v=0}^{V-1}\sum_{u=0}^{U-1}I(u,v)}$$The vertical distribution $\sigma_{v}$ of approximate electromagnetic energy:(12)$$\sigma_{v}=\sqrt{\frac{\sum_{v=0}^{V-1}\sum_{u=0}^{N-1}\left\{I(u,v)\times {\left(v-v_{c}\right)}^{2}\right\}}{\sum_{v=0}^{V-1}\sum_{u=0}^{U-1}I(u,v)}}$$


As a result, any image matrix such as $\left\{I\left(u,v\right),u\in \left[0,U-1\right],v\in \left[0,V-1\right]\right\}$ could be represented by feature vector F as follows:(13)$$F=[I_{a},\sigma_{I},\sigma_{u},\sigma_{v}]$$

In time domain, if image sample *j* in set $S_{k}$ has six image matrices, its feature vector $F_{j}$ will contain 24 feature values.

## 7. Feature Extraction from Sample Image in Frequency-Domain

In mathematics, a very important discovery was Fourier Transform which tells us that any signal is the (finite or infinite) sum of sine functions. In engineering, one of the greatest inventors was Nikolas Tesla who told us that the secret of the universe could be understood by simply thinking in terms of energy, vibration, and frequency. Such statement explicitly advises us to look for feature space and feature vector in frequency domain if we would like to understand the secret of machine intelligence.

Given image matrix $\left\{I\left(u,v\right),u\in \left[0,U-1\right],v\in \left[0,V-1\right]\right\}$, it could be represented by, or decomposed into, its Fourier series in terms of complex exponentials $e^{\pm ix}$ (in which $i=\sqrt{-1}$) which could be computed as follows:(14)$$I\left(u,v\right)=\frac{1}{UV}\sum_{\omega_{v}=0}^{V-1}\sum_{\omega_{u}=0}^{U-1}\hat{I}\left(\omega_{u},\omega_{v}\right)e^{i\left(\frac{2\pi \omega_{u}u}{U}\right)}e^{i\left(\frac{2\pi \omega_{v}v}{V}\right)}$$
with 0≤u≤U−1, 0≤v≤V−1 and:(15)$$\hat{I}\left(\omega_{u},\omega_{v}\right)=\sum_{v=0}^{V-1}\sum_{u=0}^{U-1}I(u,v)e^{-i(\frac{2\pi \omega_{u}u}{U})}e^{-i(\frac{2\pi \omega_{v}v}{V})}$$
in which $0\leq \omega_{u}\leq U-1$ and $0\leq \omega_{v}\leq V-1$. With continuous signals or data, Equation (14) will become inverse Fourier Transform while Equation (15) will become forward Fourier transform.

It is interesting to take note that each value $\hat{I}(\omega_{u},\omega_{v})$ in Equation (15) is a complex number or more precisely a vector. Mathematically speaking, a vector indicates a position in a space. Hence, Fourier coefficient vectors (or complex numbers), which are stored inside complex matrix $\left\{\hat{I}\left(\omega_{u},\omega_{v}\right),\omega_{u}\in \left[0,U-1\right],\omega_{v}\in \left[0,V-1\right]\right\}$, nicely define a feature space. Such a feature space could be called as Fourier feature space.

In mathematics, complex matrix $\left\{\hat{I}\left(\omega_{u},\omega_{v}\right),\omega_{u}\in \left[0,U-1\right],\omega_{v}\in \left[0,V-1\right]\right\}$ could be split into two ordinary matrices $\left\{A\left(\omega_{u},\omega_{v}\right),\omega_{u}\in \left[0,U-1\right],\omega_{v}\in \left[0,V-1\right]\right\}$ and $\left\{B\left(\omega_{u},\omega_{v}\right),\omega_{u}\in \left[0,U-1\right],\omega_{v}\in \left[0,V-1\right]\right\}$, where $A(\omega_{u},\omega_{v})$ is the real part of complex number (or vector) $\hat{I}(\omega_{u},\omega_{v})$ and $B(\omega_{u},\omega_{v})$ is the imaginary part of complex number (or vector) $\hat{I}(\omega_{u},\omega_{v})$. Both matrices A and B are Fourier coefficient matrices.

Therefore, in frequency domain, a straightforward way of determining feature vector F which characterizes image matrix $\left\{I\left(u,v\right),u\in \left[0,U-1\right],v\in \left[0,V-1\right]\right\}$ taken from image sample *j* in set $S_{k}$ is to re-arrange the corresponding Fourier coefficient matrix $\left\{A\left(\omega_{u},\omega_{v}\right),\omega_{u}\in \left[0,U-1\right],\omega_{v}\in \left[0,V-1\right]\right\}$ or $\left\{B\left(\omega_{u},\omega_{v}\right),\omega_{u}\in \left[0,U-1\right],\omega_{v}\in \left[0,V-1\right]\right\}$ into a vector.

Alternatively, we could use equations, which are the same to Equations (7)–(12), to compute the mean values $\left(\omega_{u,c},\omega_{v,c}\right)$ of frequencies and their standard deviations $\left(\sigma_{\omega ,u},\sigma_{\omega ,v}\right)$ from Fourier coefficient matrix $\left\{A\left(\omega_{u},\omega_{v}\right),\omega_{u}\in \left[0,U-1\right],\omega_{v}\in \left[0,V-1\right]\right\}$ or $\left\{B\left(\omega_{u},\omega_{v}\right),\omega_{u}\in \left[0,U-1\right],\omega_{v}\in \left[0,V-1\right]\right\}$. In this way, frequency domain’s feature vector corresponding to each Fourier coefficient matrix $\left\{A\left(u,v\right),u\in \left[0,U-1\right],v\in \left[0,V-1\right]\right\}$ or $\left\{B\left(\omega_{u},\omega_{v}\right),\omega_{u}\in \left[0,U-1\right],\omega_{v}\in \left[0,V-1\right]\right\}$ could be as follows:(16)$$F_{j}=[\omega_{u,c},\omega_{v,c},\sigma_{\omega ,u},\sigma_{\omega ,v}]$$

In time domain, each image sample *j* in set $S_{k}$ has three color component images. Each color component image could yield two Fourier coefficient matrices. In total, there will be six Fourier coefficient matrices for any given image sample *j* in set $S_{k}$. As a result, in frequency domain, feature vector $F_{j}$ of image sample *j* in set $S_{k}$ will also contain 24 feature values.

## 8. Cognition Process Using RCE Neural Network

Today, many researchers still believe that our mind arises from our brain. This opinion makes a lot of people or young researchers believe that the blueprint of mind is part of the blueprint of brain. For those who are familiar with microprocessors and operating systems, it is clear to us that the blueprints of operating systems are not part of the blueprints of microprocessors. 

Here, we advocate the truth which states that mind is mind while brain is brain. Most importantly, the basic functions of brain are to support memorizations and computations which are intended by mind. With this truth in mind, the future research in artificial intelligence or machine intelligence should be focused on the physical principles behind the design of human-like minds which could transform signals into the cognitive states of knowing the conceptual meanings behind the signals. 

In the previous sections, we have discussed the details of feature extraction. The results are lists of feature vectors in time domain, frequency domain, or both. Then, the next question will be how to learn the conceptual meaning behind a set of feature vectors corresponding to the same class of sample images or the same identity of sample images. The good news is that RCE neural network discovered in 1970s provides us a better version of answers, so far. 

As shown in Figure 6, both cognition and recognition could be implemented with the use of RCE neural network which consists of three layers. There is a single vector at the input layer. Also, there is a single vector at the output layer. However, inside the middle layer, there is a dynamically growing number of nodes, each of which memorizes the feature vectors from a set of sample images provided by a training session of cognition. 

Clearly, RCE neural network is fundamentally different from the so-called artificial neural network which is simply a graphical representation of a system of equations with coefficients to be tuned in some simple or deep manners (e.g., back-propagation method).

Refer to Figure 6. With training session i’s feature vectors, we could easily compute the mean vector and the standard deviation of the distances from the training session’s feature vectors to their mean vector. 

For example, if training session i has $k_{i}$ sample images which form the following set:(17)$$S_{k_{i}}=\left\{I_{j,r}\left(u,v\right),I_{j,g}\left(u,v\right),I_{j,b}\left(u,v\right),u\in \left[0,U-1\right],v\in \left[0,V-1\right],j\in \left[1,k_{i}\right]\right\}$$
then training session i’s set of feature vectors computed by feature extraction module could be denoted by $\left\{F_{i,j},j\in [1,k_{i}]\right\}$ where j is the index of image sample j in set $S_{k_{i}}$. Subsequently, the mean vector of $\left\{F_{i,j},j\in [1,k_{i}]\right\}$ could be calculated by:(18)$$F_{i,a}=\frac{1}{k_{i}}\sum_{j=1}^{k_{i}}F_{i,j}$$
and the standard deviation of the distances from $\left\{F_{i,j},j\in [1,k_{i}]\right\}$ to the mean vector could be computed by:(19)$$\sigma_{i}=\sqrt{\frac{1}{k_{i}}\sum_{j=1}^{k_{i}}{\left(F_{i,j}-F_{i,a}\right)}^{T}(F_{i,j}-F_{i,a})}$$

By now, we could explain he physical meaning of node i (i.e., outcome of training session i) in RCE neural network, which is simply the representation of hyper-sphere [21] with its center at $F_{i,a}$ and its radius to be equal to $3\sigma_{i}$. Since i could dynamically grow, RCE neural network naturally supports the process of incrementally learning as well as the process of deep learning which is widely discussed about in the literature. 

As we mentioned above, the deep tuning of parameters inside a complex artificial neural network, which is a graphical representation of a system of equations, has nothing to do with deep learning, and the true nature of deep learning is outlined in Figure 6.

In summary, a training session for cognizing entity *n* consists of supplying a set of entity *n*’s image samples in Equation (17) and entity *n*’s conceptual meaning $L_{n}$ which is a label or a word in a natural language such as English. 

## 9. Recognition Process Using Possibility Function

Refer to Figure 6 again. With a trained RCE neural network by a cognition process for each entity of interest (e.g., entity *n*), the output layer is primarily for the purpose of executing recognition process when the feature vector computed from any arbitrary image sample is given to the input layer.

In literature, many researchers believe that recognition is a process of determining the chances of occurrences. As a result, probability functions are widely used inside a recognition module. 

Here, we advocate the truth which states that recognition is a process of evaluating the beliefs about the identities and categories of any arbitrary image sample at input. This truth is in line with the fact that our mind consists of many sub-systems of beliefs. Hence, the function for estimating the degrees of beliefs should be a probability function such as:(20)$$p_{j,n}\left(i\right)=e^{-\frac{1}{2\sigma_{i}^{2}}{\left(F_{0,j}-F_{i,a}\right)}^{T}(F_{0,j}-F_{i,a})}$$
where $\left(F_{i,a},\sigma_{i}\right)$ is the parameter vector of the hyper-sphere obtained from training session i while $F_{0,j}$ is the feature vector computed from image sample j during recognition process, and $p_{j,n}(i)$ is the possibility for image sample j to belong to learnt entity n according to training session i’s parameter vector.

Since RCE neural network intrinsically supports incremental learning as well as deep learning, the single node in the output layer must include a Soft-Max function such as:(21)$$P_{j,n}=\underset{i}{\max}\left\{p_{min},p_{j,n}(i)\right\}$$
where $p_{min}$ is the minimum value of acceptable possibility (e.g., 0.5). In practice, if $P_{j,n}=p_{min}$, the interpretation could be stated as follows: input $F_{0,j}$ does not support the belief that image sample j belongs to learnt entity n. Otherwise, if $P_{j,n}>p_{min}$, it means that the output of recognition will be $\left(L_{n},P_{j,n}\right)$ in which $L_{n}$ is the conceptual meanings of image sample j.

## 10. Forward/Inverse Processes of Template Matching

In the previous sections, we have discussed the key details about the modules of image sampling, hybrid feature extraction, cognition, and recognition. In a human-like stereovision system, these modules will produce the output of recognized entities inside left camera’s image plane, as illustrated by Figure 4. Then, the next question will be how to determine the match in right camera if a recognized entity in left camera is given. This question describes the famous problem of stereo matching faced by today’s stereovision systems. 

In literature, stereo matching is a widely investigated problem. So far, there is no solution which could achieve the performance close to, or as good as, the one of human being’s stereovision system. Hence, better solutions for improved performance are still expected from future research works in this area.

In this paper, we present a new strategy which could cope with the problem of stereo matching in a better way. This new strategy consists of the interplay between forward template matching and inverse template matching.

In stereovision, the only geometrical constraint is the so-called epipolar line which indicates the possible locations of a match (e.g., at location b in Figure 7) in right image plane if a location in left image plane is given (e.g., location a in Figure 7). 

As shown in Figure 7, if recognized sample j at location a is given in left image plane, the forward process of template matching will consist of the following steps:Determine the equation of epipolar line from both stereovision’s calibration parameters (NOTE: such knowledge could be found in any textbook of computer vision) and location a’s coordinates.Scan the epipolar line location by location.Take image sample e at currently scanned location *e*.Compute the feature vector of image sample e.Compute the cosine distance between image sample j’s feature vector and image sample e’s feature vector.Repeat the scanning until it is completed.Choose the image sample to be the candidate of matched sample j’ if it minimizes the cosine distance.Use the cosine distance between recognized sample j and the chosen candidate of matched sample j’ to compute the possibility value of match (i.e., to use Equation (20)).Accept matched sample j’ if the possibility value of match is greater than a chosen threshold value (e.g., 0.5).


In the above process, if $F_{j}$ is the feature vector of image sample j while $F_{e}$ is the feature vector of image sample e, the cosine distance $d_{j,e}$ between those two vectors is simply calculated according to their inner product, which is:(22)$$d_{j,e}=\frac{F_{j}^{T}\times F_{e}}{\left\|F_{j}\right\|\times \left\|F_{e}\right\|}$$
and the corresponding possibility value is calculated as follows:(23)$$P_{j,e}=e^{-\frac{1}{2\sigma_{0}^{2}}d_{j,e}^{2}}$$
where $\sigma_{0}$ is a default value of standard deviation which could be self-determined by robots, vehicles, or machines during a training session of cognition process.

According to the illustration shown in Figure 3, the forward process of template matching will work only if there is no partial view due to either occlusion or image sampling. If matched sample j’ in right image contains partial view of recognized sample j in left image, the inverse process of template matching will perform better than its counterpart of forward process. 

As shown in Figure 8, if recognized sample j at location a is given in left image plane, the inverse process of template matching consists of the following steps:Determine the equation of epipolar line from both the stereovision’s calibration parameters and the location a’s coordinates.Scan the epipolar line location by location.Take image sample e at currently scanned location e.Divide image sample e into a matrix of sub-samples $\left\{e_{i},i=1,2,3,\ldots \right\}$.Use each sub-sample in $\left\{e_{i},i=1,2,3,\ldots \right\}$ as template and do forward template matching with recognized sample j.Compute the mean value of all the possibility values which measure the match between all the sub-samples in $\left\{e_{i},i=1,2,3,\ldots \right\}$ and recognized sample j. This mean value represents the possibility value for image sample e in right image to match with recognized sample j in left image.Repeat the scanning until it is completed.Choose the image sample to be the candidate of matched sample j’ if it minimizes the possibility values of match (i.e., calculated by Equation (23)).Accept the match if the possibility value of match is greater than a chosen threshold value (e.g., 0.5).


In practice, we could run both forward process and inverse process of template matching in parallel. In this way, a better decision of match in right image could be made if recognized sample j in left image is given.

## 11. Implementation and Results

The proposed new principle has been implemented in Python. Preliminary tests have been with image data from public domain. Especially, we use image data which are posted to public domain by maritime RobotX challenge (www.robotx.org (accessed on 15 March 2023)). Figure 9 shows two typical examples of scenes constructed by maritime RobotX challenge. 

The tasks to be undertaken by an autonomous surface vehicle include stereovision-guided delivery of objects, stereovision-guided parking into the docking bay, etc. In the following sections, we share some of our experimental results.

### 11.1. Results of Top-Down Sampling Strategy of Input Images

Our proposed top-down sampling strategy of input images (e.g., images from left camera) is to divide an input image from left camera into a list of sets $S_{k}$ which contain increasing number of image samples (i.e., *k* = 1, 2, 3, …). Figure 10 shows an example of results (i.e., *k* = 28) from our proposed top-down sampling strategy of an input image. At this level of sampling, a red floating post clearly appears inside one of these 28 samples.

### 11.2. Examples of Training Data for Cognition (i.e., Learning)

The proposed new principle involves the use of cognition and recognition modules. For cognition module, it is necessary to train it with training data of reference entities. Without loss of generality, we simply use a set of 10 samples to train the cognition module which is specifically dedicated to an entity of interest. It is amazing to see that the proposed solution could achieve successful results with 10 samples inside a dataset of training for each entity of interest.

Figure 11 shows the scenario of autonomous parking into a docking bay by an autonomous surface vehicle. In this task, the mental capabilities of the autonomous surface vehicle include (a) cognition of triangle, cross, and circle, and (b) recognition of triangle, cross and circle. Hence, for the training of cognition module dedicated to each entity among triangle, cross and circle, we simply take ten samples as shown in Figure 11.

### 11.3. Results of Feature Extraction in Time Domain

For each sample image in Figure 10, we calculate its feature vector in time domain. Here, we share the results of feature vectors computed from the ten image samples of triangle in Figure 11. These results are shown in Figure 12, which also gives the result of the mean vector and its standard deviation.

### 11.4. Results of Feature Extraction in Frequency Domain

For each sample image in Figure 10, we calculate its feature vector in frequency domain. Similarly, we share the results of feature vectors computed from the ten image samples of triangle in Figure 11. These results are shown in Figure 13, which also gives the result of the mean vector and its standard deviation.

### 11.5. Results of Cognition

The mean vector and its standard deviation, which are obtained from each training session for any entity of interest, will be stored inside a node at the middle layer of the RCE neural network which is allocated to an entity’s cognition module. If there are total number N of entities of interest, there will be a set of N RCE neural networks which support N pairs of cognizers and recognizers such as {(Cognizer n Recognizer n), n = 1, 2, 3, …, N} (i.e., n refers to the n-th entity). As illustrated in Figure 14, the total number N of entities could incrementally grow very deeply.

### 11.6. Results of Recognition

With total number N of pairs of cognizers and recognizers such as {(Cognizer n, Recognizer n), n = 1, 2, 3, …, N} in place, an autonomous surface vehicle or robot is ready to recognize familiar or learnt entities inside images of left camera. 

Figure 15 shows two examples of results of recognition in time domain. Each example contains seven image samples as input. Among these seven inputs, three of them are totally out of the class dedicated to the pair of cognizer and recognizer. We can see that recognition module performs quite successfully in recognizing the correct entries. Please take note that the feature vectors of image samples at input are all in time domain. 

With the same image samples at input, Figure 16 shows the results of recognition in frequency domain. We can see that recognition module also performs quite successfully in recognizing the correct entries. Please take note that the feature vectors of image samples at input are all in frequency domain.

By now, people may ask whether the proposed new principle could work well with other more complex entities. For the sake of responding to such doubt, we give two more examples of results which make use of the feature vectors in frequency domain to do cognition and recognition. For each reference entity (e.g., car and dog), the cognition module is trained with ten image samples while the recognition process is tested with six image samples as input.

In Figure 17, we show the experimental results of cognizing and recognizing cars in frequency domain. The results are judged to be very good.

In Figure 18, we show the experimental results of cognizing and recognizing dogs in frequency domain. The results are also judged to be very good.

### 11.7. Results of Stereo Matching

Mathematically, a pair of images is good enough to validate a stereo matching algorithm. In practice, a pair of images could come from a binocular vision system which is normally named as stereovision system. Alternatively, a pair of images could come from a mobile monocular vision system. Since we use the image dataset from the public domain, it is easier for us to take two images from an image sequence captured by a mobile camera.

Here, we share one example of results in Figure 19, Figure 20 and Figure 21. In Figure 19, we let the stereovision system undergo the cognition process in which ten sample images of a floating post are used to train the RCE neural network inside the cognizer allocated to learn the red floating post. After the training of the cognizer’s RCE neural network, the so-called stereovision system is ready to enter the recognition process which takes any set of new images as input. 

In Figure 19, seven image samples are selected for testing the validity of trained RCE neural network. The possibility values show good outcome from the recognition process.

Now, we could start the process of stereo matching. As shown in Figure 20, the first step is to do image sampling. When the so-called left image is sampled into a matrix of 4 × 7 image samples, the occurrence of a red floating post could be recognized. Please take note that the image sample of this recognized occurrence is named as image sample 1a by our testing program. 

Subsequently, in the so-called right image, we could determine a line (i.e., equivalent to epipolar line) which will guide the search for the best match candidate. 

For the purpose of illustration, we take three image samples to compute the stereo matching results. Among these three image samples, image sample 1b is the best match. In general, the best match is the one which maximizes the possibility value (i.e., computed by Equation (23) with $\sigma_{0}=10$) between image sample 6a in left image and all the possible image samples in right image. Figure 21 shows the possibility values computed for three pairs of possible matches, which are (1a, 1b), (1a, 1c), and (1a, 1d). Clearly, pair (1a, 1b) stands out to be the best match.

## 12. Conclusions

In this paper, we have described the details of the key steps in a proposed new principle which aims at achieving robust stereo matching in human-like stereovision. The main idea is to undertake stereo matching at a cognitive level. The significant contributions from this paper include: First, the introduction of a top-down sampling strategy will lighten the burden of subsequent processes in stereovision. This is because it will provide better versions of image samples, which will in return diminish the chance of committing errors by the subsequent processes in stereovision. Secondly, we advocate the process of feature extraction in both time domain and frequency domain. In this way, key characteristics of a visual entity will be able to be preserved as much as possible. Especially, we have highlighted the importance of Fourier series and Fourier coefficients in the process of extracting visual features from images. Thirdly, we have shown the important difference between artificial neural network and RCE neural network. Most importantly, we have introduced the possibility function to improve RCE neural network so as to make it a better way to support the process of cognition (including deep learning) as well as the process of recognition. Fourthly, we have introduced the inverse strategy of template matching. This is a better solution to cope with the problem of partial views due to occlusions or mis-aligned sampling of images. Finally, the key steps in the proposed new principle have been validated by experiments with real image data under the context of maritime RobotX challenge. The obtained results are very encouraging. We hope that more results and progress will emerge in this new direction of research.

Last, but not least, it is important to highlight the advantages and limitations of the proposed new principle. The notable advantages include: (a) fast responses which are independent of the left camera’s resolution because of the top-down sampling strategy, (b) fast attentions which focus on entities of interest according to the decreasing order of entities’ sizes, (c) computational efficiency because of cognition process which only runs on images from one camera, and (d) meaningful outputs in terms of what, who and where at the same time. The notable limitation of the proposed new principle is the prerequisite of training (i.e., cognition) which makes the stereovision systems to only respond to familiar entities. Future works include at least: (a) to test the robustness in the presence of partial views due to occlusions or mis-aligned image samples, (b) to explore more ways of doing the extraction of hybrid features, (c) to collect more training data of more entities of interest, (d) to extend the works to the case of stereovision for humanoid robots, and (e) to venture into the situation of cognizing and recognizing sub-entities. 

## Figures and Tables

**Figure 1 biomimetics-08-00285-f001:**
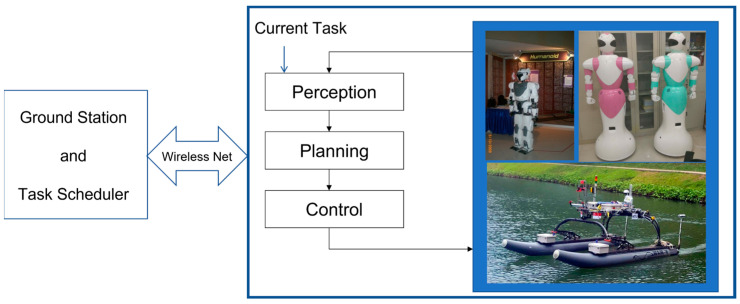
Research Framework Underlying the Development of Intelligent Humanoid Robots and Autonomous Surface Vehicles.

**Figure 2 biomimetics-08-00285-f002:**
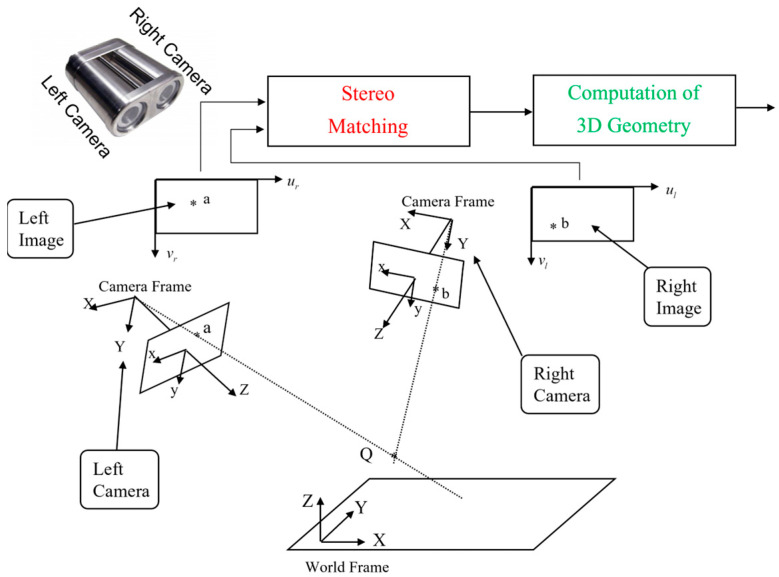
Illustration of Stereo Matching. “*” indicates the location of entity in 3D scene as well as its locations in both left image plane and right image plane.

**Figure 3 biomimetics-08-00285-f003:**
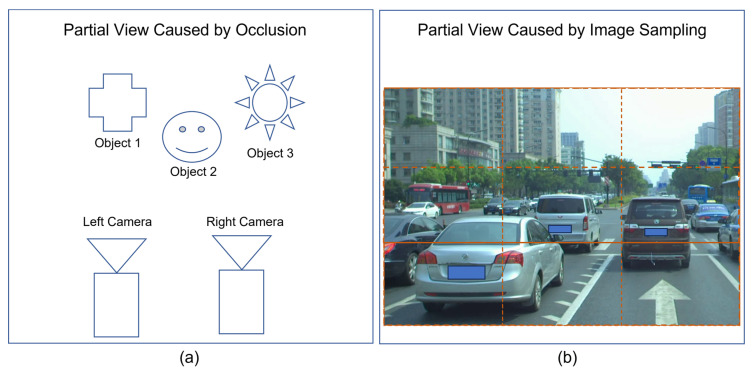
Illustration of (**a**) partial view caused by occlusion in which left camera sees partial view of object 3 while right camera sees partial view of object 1, and (**b**) partial view caused by image sampling in which the three nearest vehicles partially appear inside samples at row 2 and row 3.

**Figure 4 biomimetics-08-00285-f004:**
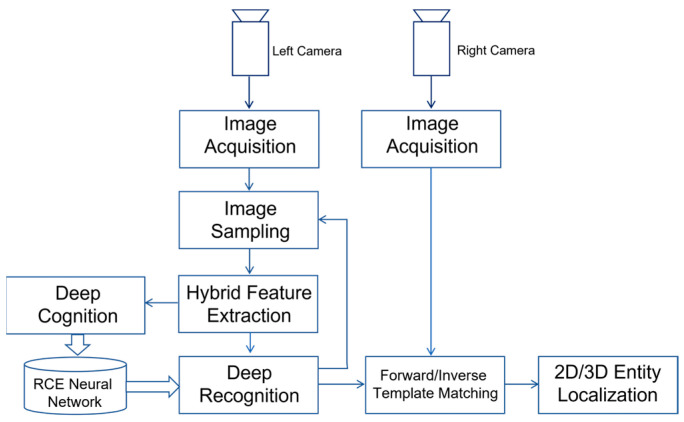
Outline of Proposed New Principle Toward Achieving Robust Matching in Human-like Stereovision.

**Figure 5 biomimetics-08-00285-f005:**
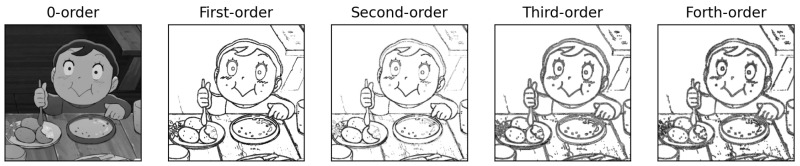
Examples of results from the computations of zero-order derivatives, first-order derivatives, second-order derivatives, third-order derivatives, and forth-order derivatives.

**Figure 6 biomimetics-08-00285-f006:**
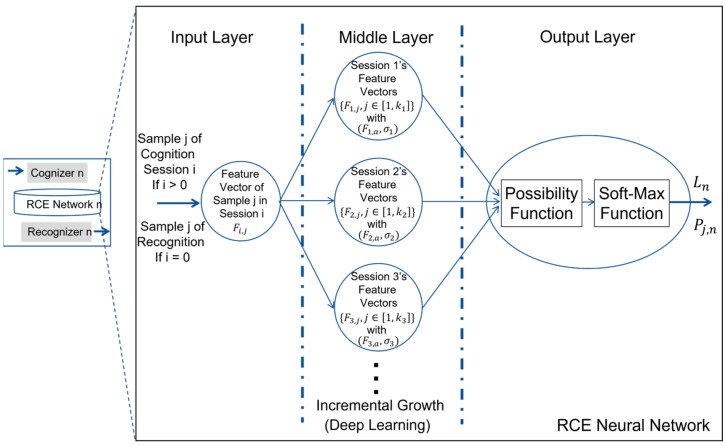
Structure of RCE Neural Network.

**Figure 7 biomimetics-08-00285-f007:**
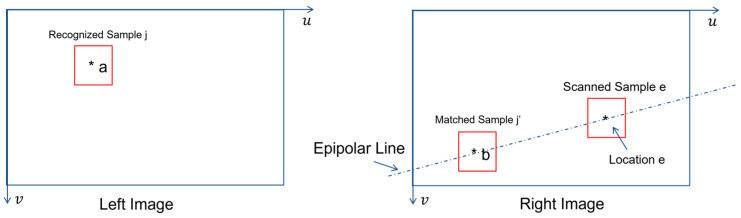
Illustration of Forward Template Matching in Stereovision.

**Figure 8 biomimetics-08-00285-f008:**
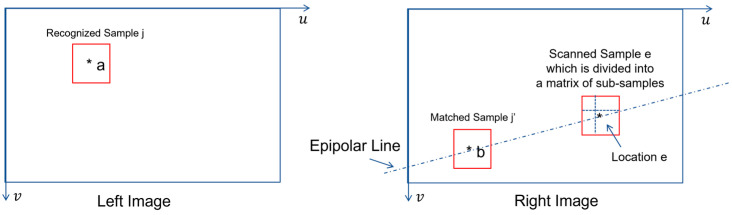
Illustration of Inverse Template Matching in Stereovision.

**Figure 9 biomimetics-08-00285-f009:**
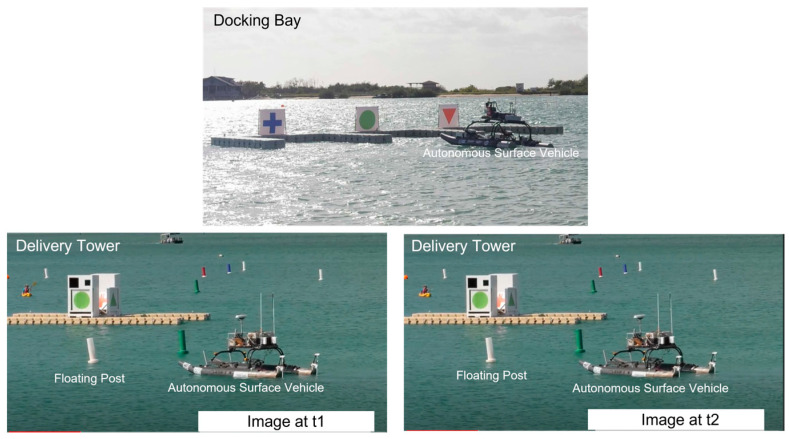
Typical scenes constructed by maritime RobotX challenge.

**Figure 10 biomimetics-08-00285-f010:**
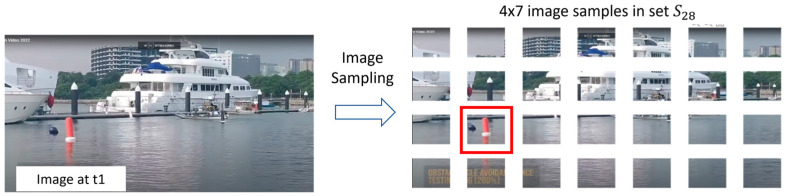
One Result of Top-down Sampling Strategy of Input Image.

**Figure 11 biomimetics-08-00285-f011:**
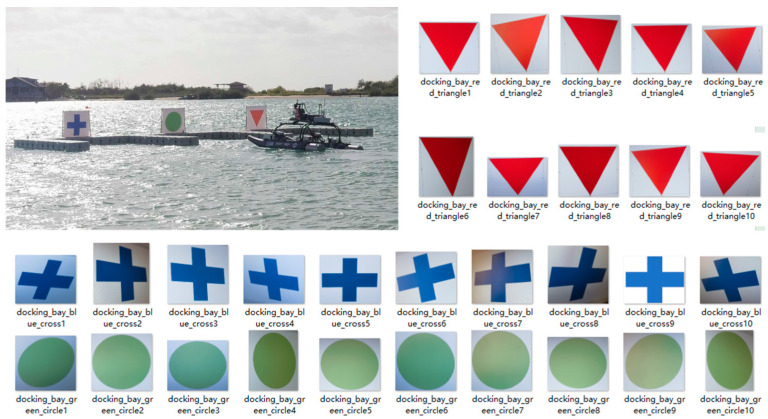
Ten Sample Images for Training Cognition Module Dedicated to Each Entity Among Triangle, Cross and Circle.

**Figure 12 biomimetics-08-00285-f012:**
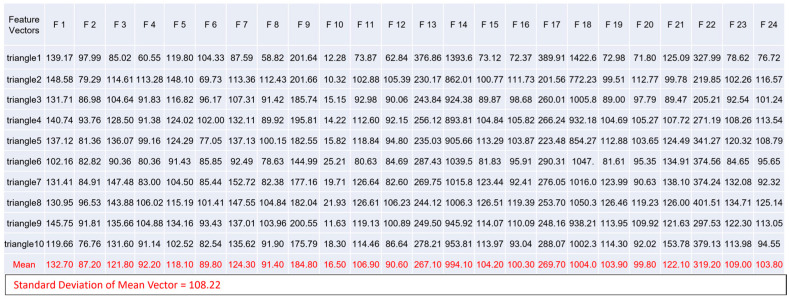
The Values of Ten Feature Vectors Computed from Ten Sample Images of Triangle in Time Domain. The statistics computed from these 10 feature vectors are highlighted in red color.

**Figure 13 biomimetics-08-00285-f013:**
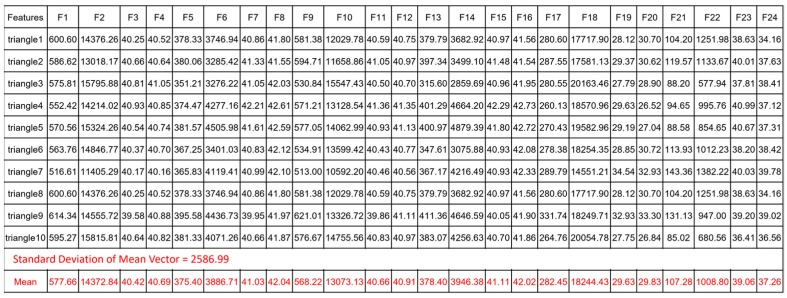
The Values of Ten Feature Vectors Computed from Ten Sample Images of Triangle in Frequency Domain. The statistics computed from these 10 feature vectors are highlighted in red color.

**Figure 14 biomimetics-08-00285-f014:**
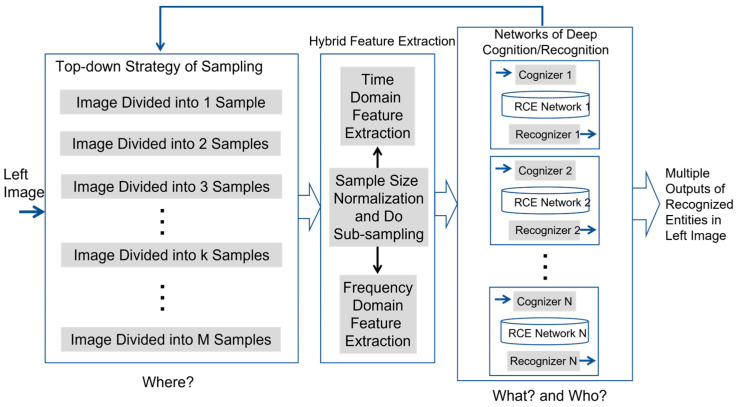
Results of Cognition in the Form of total number N of Cognizers (i.e., 1, 2, 3, …, N).

**Figure 15 biomimetics-08-00285-f015:**
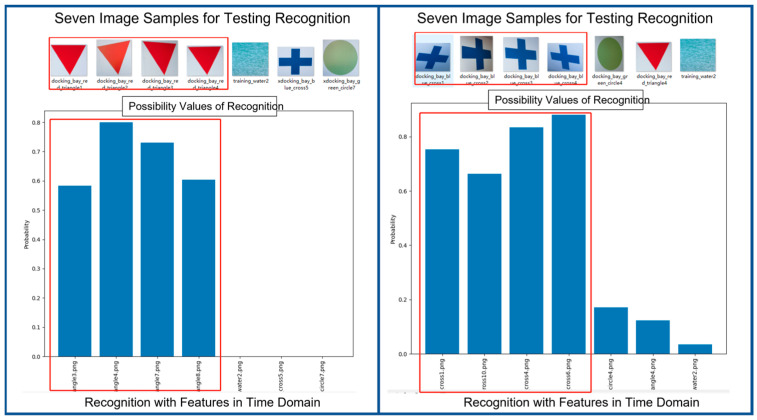
Two Examples of Results of Recognition Using Feature Vectors in Time Domain.

**Figure 16 biomimetics-08-00285-f016:**
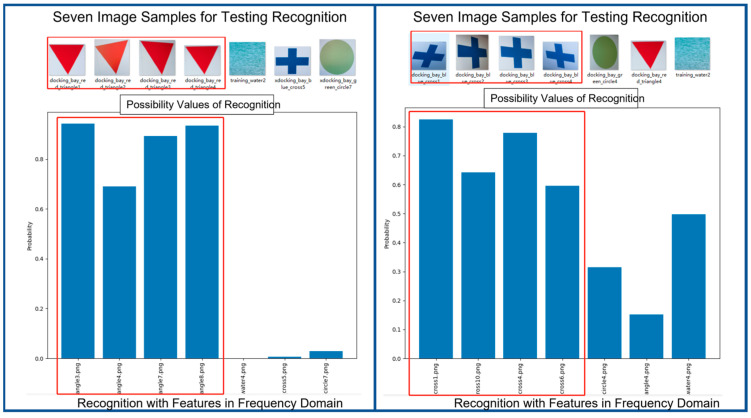
Two Examples of Results of Recognition Using Feature Vectors in Frequency Domain.

**Figure 17 biomimetics-08-00285-f017:**
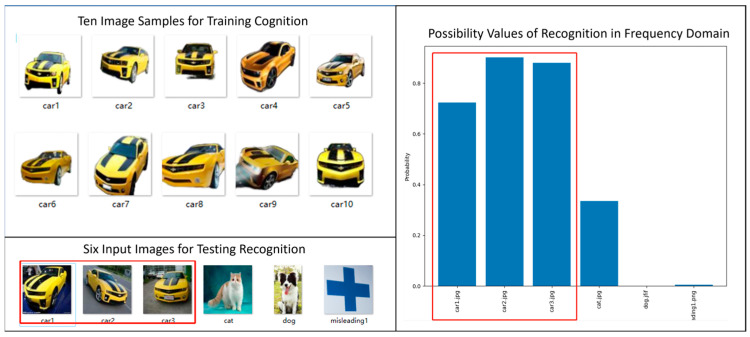
Example of Cognizing and Recognizing Cars in Frequency Domain.

**Figure 18 biomimetics-08-00285-f018:**
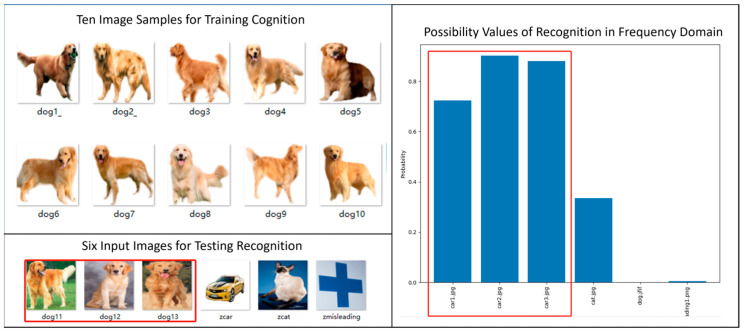
Example of Cognizing and Recognizing Dogs in Frequency Domain.

**Figure 19 biomimetics-08-00285-f019:**
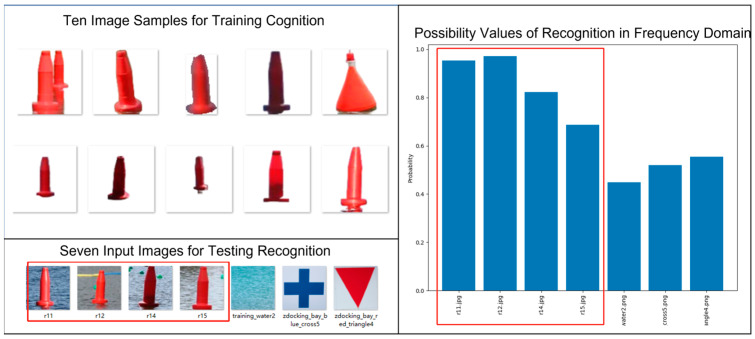
Results of Cognizing and Recognizing Red Floating Posts in Frequency Domain.

**Figure 20 biomimetics-08-00285-f020:**
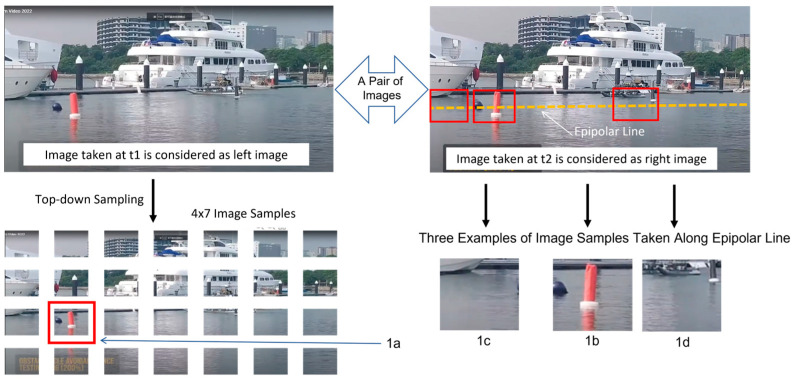
Results of Testing Recognition with Seven Image Samples, after Doing Cognition with Ten Image Samples Which Have Certain Level of Intended Variations for the Purpose of Appreciating Robustness. The image sample recognized in left image is denoted “1a” while three matching candidates of image samples in right image are denoted “1c”, “1b”, and “1d”, respectively.

**Figure 21 biomimetics-08-00285-f021:**
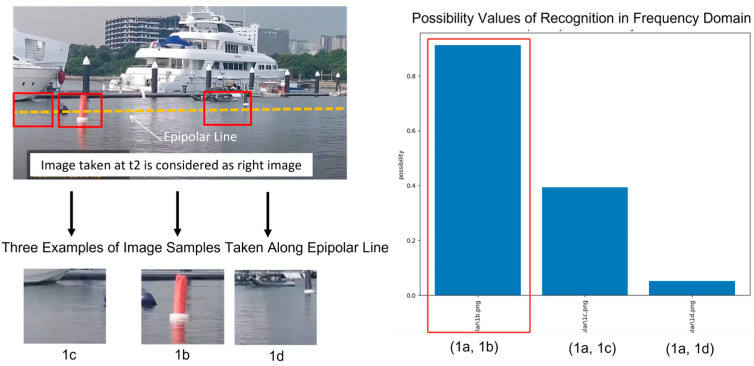
Results of Stereo Matching Among Three Possible Pairs of Matches Which Are: pair(1a, 1b), pair(1a, 1c), and pair(1a, 1d).

## Data Availability

All data related to this paper will be made available upon request. The access to the data is subject to the data protection policy of Nanyang Technological University, Singapore.

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
