# Peer review of "A New Principle toward Robust Matching in Human-like Stereovision"

_biomimetics, 2023, doi:10.3390/biomimetics8030285_

Round 1

Reviewer 1 Report

In this paper,  the details of a new principle toward achieving a robust matching solution which leverages on the use and integration of top-down image sampling strategy, hybrid feature extraction, and RCE neural network for incremental learning (i.e., cognition) as well as robust match-maker (i.e., recognition) are presented.

This paper should be accepted subject to the following conditions:

1.       Each abbreviation has to be explained is the paper. Please explain abbreviation RCE which is mentioned in the Abstract of the paper.

2.       Limitations of existing methods for solving g the problem faced by stereo matching have to be mentioned in the paper.

3.       The advantages of the developed new principle toward achieving a robust matching solution have to be discussed in the paper.

4.       Plans for future work have to be mentioned in the Conclusion.

Author Response

The authors would like to express their sincere thanks to the reviewer for giving valuable suggestions and feedback which help to improve the manuscript.

Below are our replies to the reviewer’s suggestions and feedback:

  1. Each abbreviation has to be explained is the paper. Please explain abbreviation RCE which is mentioned in the Abstract of the paper.

[The full name is given before its first usage in the Abstract]

  1. Limitations of existing methods for solving the problem faced by stereo matching have to be mentioned in the paper.

[Added in Conclusions]

  1. The advantages of the developed new principle toward achieving a robust matching solution have to be discussed in the paper.

[Added in Conclusions]

  1. Plans for future work have to be mentioned in the Conclusion.

[Added in Conclusions]

Reviewer 2 Report

1.In figures 1, 3(a), and 14, attention should be paid to beautification, and text should not overlap with the image. Also, the positions of the three black dots should be adjusted.

2.The annotations for figure 2 should be on the same page as figure 2.

3.In formula 7, Ia is not introduced in the discussion. What does it represent?

4.In section 11.1, if the red floating post does not appear exactly in the sampling block, but is distributed in two different sampling blocks, how should it be matched? Can the method proposed in the article effectively identify and match it?

5.In the detection result image of figure 18, why is the recognition rate for dogs 0, but still considered a very good result?

Author Response

The authors would like to express their sincere thanks to the reviewer for giving valuable suggestions and feedback which help to improve the manuscript.

Below are our replies to the reviewer’s suggestions and feedback:

  1. In figures 1, 3(a), and 14, attention should be paid to beautification, and text should not overlap with the image. Also, the positions of the three black dots should be adjusted.

[Rectified.]

2.The annotations for figure 2 should be on the same page as figure 2.

[Rectified.]

  1. In formula 7, Ia is not introduced in the discussion. What does it represent?

[Amended in page 7 and page 8.]

  1. In section 11.1, if the red floating post does not appear exactly in the sampling block, but is distributed in two different sampling blocks, how should it be matched? Can the method proposed in the article effectively identify and match it?

[Good question. In this paper, we did not focus too much on cases in which partial views occur. Future works will include the test on the application of inverse template matching to cope with partial views due to mis-aligned image samples. This content has been added to the Conclusions.]

  1. In the detection result image of figure 18, why is the recognition rate for dogs 0, but still considered a very good result?

[In Figure18, there are six image samples for testing recognition.

The first three of them are images of dogs.

The fourth image is a car.

The fifth image is a cat, but not dog.

The sixth image is a cross sign.

The recognition rates for the three images of dogs are very high (far above 50%) while the recognition rates for the remaining three images are very low (all are below 50%).]
